# Arrhythmia-Associated Calmodulin E105A Mutation Alters the Binding Affinity of CaM to a Ryanodine Receptor 2 CaM-Binding Pocket

**DOI:** 10.3390/ijms242115630

**Published:** 2023-10-26

**Authors:** Angelos Thanassoulas, Maria Theodoridou, Laila Barrak, Emna Riguene, Tamader Alyaarabi, Mohamed A. Elrayess, F. Anthony Lai, Michail Nomikos

**Affiliations:** 1College of Medicine, QU Health, Qatar University, Doha P.O. Box 2713, Qatar; athanasoulas@qu.edu.qa (A.T.); laila.barrak@yahoo.fr (L.B.); emna.riguene@qu.edu.qa (E.R.); ta1701499@student.qu.edu.qa (T.A.); m.elrayess@qu.edu.qa (M.A.E.); tonylaicf103ax@gmail.com (F.A.L.); 2Biomedical Research Center, Qatar University, Doha P.O. Box 2713, Qatar; ma-theodo@hotmail.com

**Keywords:** calmodulin, ryanodine receptor, RyR2, arrhythmias, cardiac disease

## Abstract

Calmodulin (CaM) is a small, multifunctional calcium (Ca^2+^)-binding sensor that binds and regulates the open probability of cardiac ryanodine receptor 2 (RyR2) at both low and high cytosolic Ca^2+^ concentrations. Recent isothermal titration calorimetry (ITC) studies of a number of peptides that correspond to different regions of human RyR2 showed that two regions of human RyR2 (3584-3602aa and 4255-4271aa) bind with high affinity to CaM, suggesting that these two regions might contribute to a putative RyR2 intra-subunit CaM-binding pocket. Moreover, a previously characterized de novo long QT syndrome (LQTS)-associated missense CaM mutation (E105A) which was identified in a 6-year-old boy, who experienced an aborted first episode of cardiac arrest revealed that this mutation dysregulates normal cardiac function in zebrafish by a complex mechanism that involves alterations in both CaM-Ca^2+^ and CaM-RyR2 interactions. Herein, to gain further insight into how the CaM E105A mutation leads to severe cardiac arrhythmia, we generated large quantities of recombinant CaM^WT^ and CaM^E105A^ proteins. We then performed ITC experiments to investigate and compare the interactions of CaM^WT^ and CaM^E105A^ mutant protein with two synthetic peptides that correspond to the two aforementioned human RyR2 regions, which we have proposed to contribute to the RyR2 CaM-binding pocket. Our data reveal that the E105A mutation has a significant negative effect on the interaction of CaM with both RyR2 regions in the presence and absence of Ca^2+^, highlighting the potential contribution of these two human RyR2 regions to an RyR2 CaM-binding pocket, which may be essential for physiological CaM/RyR2 association and thus channel regulation.

## 1. Introduction

Ryanodine receptor type 2 (RyR2) is a massive (~2.2 MDa) tetrameric protein that controls the calcium (Ca^2+^) release from the sarcoplasmic reticulum (SR), playing a critical role in excitation–contraction coupling (ECC) in cardiac muscle [1,2]. RyR2-mediated Ca^2+^ release is finely orchestrated by an array of synchronized bidirectional interactions with various membrane-bound, cytoplasmic, and sarcoplasmic lumen proteins. Ca^2+^ entry into the cell, mediated by the L-type voltage-gated Ca^2+^ channel (Ca_v_1.2), is the physiological trigger for the opening of the RyR2 channel, a process commonly referred to as calcium-induced calcium release (CICR) [3,4]. Mutations in RyR2 (over 150 have been reported up to date) have been associated with life-threatening arrhythmogenic disorders, such as catecholaminergic polymorphic ventricular tachycardia (CPVT), idiopathic ventricular fibrillation (IVF) and arrhythmogenic right ventricular dysplasia type 2 that can lead to sudden cardiac death [4,5,6,7,8,9]. Due to its vital role in cardiac muscle contraction, the RyR2 channel has to be tightly regulated to ensure proper cardiac muscle function. A large number of RyR2 modulators has been reported over the last decades. These include monovalent and divalent cations, particularly Ca^2+^, which is the main physiological RyR2 modulator from both cytosolic and luminal sites, other small molecules (such as ATP and caffeine), as well as other accessory proteins, including FK506-binding protein 12 and 12.6 (FKBP12/12.6) and calmodulin (CaM) [4,10,11,12,13,14,15,16].

Numerous studies have demonstrated the importance of RyR2 regulation by CaM, highlighting its critical role in normal cardiac function [15,16,17]. CaM binds stoichiometrically to RyR2 (1 CaM per RyR2 subunit) and inhibits the open probability of the channel at both low and high cytosolic Ca^2+^ concentrations [18,19]. Despite a relatively small size (148 amino acids) and basic domain architecture, CaM is composed of two globular domains (N- and C-lobes) with two EF-hand motifs each, able to bind calcium ions with high affinity. CaM acts as an intracellular Ca^2+^ sensor mediating a wide range of vital cellular processes, including muscle contraction, neurotransmitter release, and gene expression [20]. Calmodulin can exist in two major conformations: the calcium-bound (holo) and calcium-free (apo) forms. Upon Ca^2+^ binding, CaM undergoes major conformational changes, exposing hydrophobic regions, which enable it to interact specifically with its numerous binding partners in the cell, in a Ca^2+^-dependent manner [20,21]. In humans, there are three CaM genes (*CALM1*, *CALM2*, and *CALM3*), all producing an identical protein sequence and all of them are expressed in cardiac tissue [22]. During the last decade, various clinical and genetic studies have reported over 30 CaM missense mutations in the three CaM genes, which were found in individuals with severe arrhythmogenic disorders, such as catecholaminergic polymorphic ventricular tachycardia (CPVT), long QT syndrome (LQTS), idiopathic ventricular fibrillation (IVF) and early onset sudden cardiac death [23,24,25,26,27,28]. As a result of the sophisticated and multifaceted role of CaM in the heart, the underlying molecular basis of these disorders is not yet fully understood. However, different molecular mechanisms have been proposed through which CaM missense mutations can lead to the aforementioned arrhythmogenic phenotypes. One primary mechanism involves abnormal cytosolic Ca^2+^ levels produced as a result of increased Ca^2+^ “leakage” from SR due to reduced/abolished inhibition of RyR2 [29,30,31,32,33,34,35,36,37,38]. Many studies have reported different RyR2 regions as potential CaM-binding sequences, with the sequence that lies within the residues 3583-3603aa of RyR2 as the most well-established RyR2 CaM-binding region [16,39,40,41,42]. We recently showed that CaM interacts with two peptides that correspond to two different human RyR2 CaM-binding regions (3584-3602aa and 4255-4271aa), and we proposed that these two regions might contribute to a putative intra-subunit CaM-binding pocket [38].

We have previously characterized a de novo LQTS-associated missense CaM mutation (E105A) which was identified in a 6-year-old boy, who experienced an aborted first episode of cardiac arrest [27,37]. In that study, we proposed that CaM E105A mutation can dysregulate normal cardiac function in zebrafish by a complex mechanism that involves alterations in both CaM-Ca^2+^ and CaM-RyR2 interactions [38].

The aim of the present study was to shed more light on how the CaM E105A mutation leads to this life-threatening arrhythmogenic phenotype by investigating the biomolecular impact of this mutation on CaM-RyR2 interaction. Thus, we used a bacterial expression system to produce and purify large amounts of recombinant CaM^WT^ and CaM^E105A^ proteins for isothermal titration calorimetry (ITC) experiments to compare the interactions of CaM^WT^ and CaM^E105A^ mutant proteins with the two synthetic peptides corresponding to human RyR2 regions (3584-3602aa and 4255-4271aa), which we previously proposed may contribute to a potential intra-subunit RyR2 CaM-binding pocket.

## 2. Results

### 2.1. E105A Mutation Alters the Affinity of CaM for Peptide B (RyR2 3581-3607aa) in the Presence and Absence of Ca^2+^

To gain further insight into how the CaM E105A mutation leads to severe cardiac arrhythmia and determine quantitatively how this specific mutation alters the association of CaM with RyR2 leading to diminished inhibition, we initially bacterially expressed and purified large quantities of recombinant CaM^WT^ and CaM^E105A^ proteins, as we have previously described [37]. We then performed ITC experiments to investigate and compare the interactions of CaM^WT^ and CaM^E105A^ mutant protein with two synthetic peptides that correspond to the two human RyR2 regions, which we have recently proposed to contribute to a putative intra-subunit CaM-binding pocket [38]. The first peptide (B) corresponds to the well-established RyR2 CaM-binding region (3581-3607aa), while the second peptide (F) corresponds to another putative CaM-binding region (4240-4277aa), which we previously found to interact with significant affinity with CaM, both in the presence and absence of Ca^2+^ [38]. Figure 1 shows typical ITC profiles for the interactions between peptide B and CaM^WT^ and CaM^E105^ proteins in holo-buffer, at 25 °C. The reactions of CaM^WT^ and CaM^E105^ proteins with peptide B appeared to be exothermic, with the large exothermic signals signifying the formation of a network of favorable bonds between the receptor and ligand. While both CaM^WT^ and CaM^E105^ proteins showed an affinity for peptide B, forming a 1:1 complex, the E105A mutation appeared to produce a very significant negative effect on the binding of CaM to this peptide. More specifically, CaM^E105A^ showed a much lower affinity for peptide B compared to CaM^WT^, with a K_d_ ~5.5 times higher compared to that of CaM^WT^ (K_d_ values were 2.29 vs. 0.41 μM, respectively); (Table 1). The typical ITC profiles at 25 °C for the interactions between peptide B and CaM^WT^ and CaM^E105^ proteins in apo-buffer are shown in Figure 2. Consistent with our previous findings, the apo-CaM^WT^-peptide B binding is a much weaker endothermic interaction (K_d_ = 4.54 μM) compared to holo-CaM^WT^, (Table 1), with a complex formation that is based solely on entropically favorable contributions that typically arise through hydrophobic residues at the protein–solvent interface that are shielded from water upon binding. Interestingly, the affinity of CaM^E105A^ for peptide B was also reduced in the absence of Ca^2+^, with a K_d_ value ~1.5 times higher compared to that of the CaM^WT^ protein (K_d_ = 7.04 μM).

### 2.2. E105A Mutation Alters the Affinity of CaM for Peptide F (RyR2 4255-4271aa) in the Presence and Absence of Ca^2+^

In a similar fashion, Figure 3 shows typical ITC profiles at 25 °C for the interactions between peptide F with CaM^WT^ and CaM^E105^ in holo-buffer. Both the CaM proteins interact with peptide F, each forming a 1:1 complex, however, CaM^E105^ showed a significantly lower affinity for peptide F compared to CaM^WT^, with a K_d_ value ~4 times higher than that of CaM^WT^ (2.59 vs. 0.63 μM, respectively; Table 2). Moreover, as we have reported for CaM^WT^, the binding of both CaM^WT^ and CaM^E105A^ proteins to peptide F is endothermic. Similar to the CaM-peptide B titrations in apo-buffer, CaM^WT^, and CaM^E105A^ proteins were able to interact with peptide F in apo-buffer (Figure 4). In agreement with previous observations, the apo-CaM^WT^ interaction is ~18 times weaker compared to that of the holo-CaM^WT^ with peptide F, although it follows a similar entropically driven process. The endothermic binding signature of peptide F-CaM complex indicates a binding event, which is stabilized mainly by electrostatic interactions and hydrophobic effects. In contrast, the binding of CaM to peptide B in the presence of Ca^2+^ shows an exothermic ITC thermogram indicating favorable interactions of peptide–protein polar groups. In the absence of Ca^2+^, the total charge of the protein is altered giving a binding curve that is governed mainly by electrostatic interactions. Moreover, the apo-CaM^E105A^-peptide F interaction appeared to be ~2 times weaker compared to apo-CaM^WT^-peptide F (K_d_ values 24.48 vs. 11.05 μM, respectively). These data strongly support our previous findings regarding the deleterious effect of the E105A mutation on the interaction of CaM protein with RyR2.

## 3. Discussion

Calmodulinopathies are rare but potentially fatal arrhythmia syndromes that result in diverse clinical presentations, and they are caused by mutations in any of the three *CALM* genes. The two most prevalent phenotypes for patients with CaM mutations are LQTS and CPVT, but other phenotypes such as IVF, hypertrophic cardiomyopathy, and sudden cardiac death have been reported [28,43]. More intriguingly, patients with mixed phenotypes, displaying clinical features of both LQTS and CPVT have also been identified [28]. The importance of calmodulinopathies is further highlighted by the 2019 International Calmodulin Registry, in which over 50% of the reported patients experienced recurrent cardiac events despite the different management strategies that included beta-blockers, sodium channel blockers, other antiarrhythmic drugs or mixed therapy [44,45,46]. Thus, a better understanding of the complicated molecular mechanism(s) underlying calmodulinopathies is vital for the development of novel treatment strategies in the near future.

The physiological role of CaM in cardiomyocytes is very complex and different molecular disease-causing mechanisms have been proposed for the more than 30 CaM missense mutations that have been linked to life-threatening arrhythmogenic phenotypes. One probable mechanism involves non-physiological levels of cytosolic Ca^2+^, as a result of irregular Ca^2+^ influx through the L-type Ca^2+^ channels (Ca_v_1.2), which remain in an open configuration longer than normal. Another proposed mechanism involves higher levels of cytosolic Ca^2+^ through increased SR Ca^2+^ “leakage”, due to defective inhibition of cardiac RyR2 by its regulating proteins. Many studies have used a wide range of methods to characterize pathogenic CaM mutations and delineated how these mutations lead to loss of function through defective interaction with CaM-binding partners including the cardiac ryanodine receptor type 2 (RyR2), as well as the voltage-operated potassium (K^+^), sodium (Na^+^) and L-type Ca^2+^ channels in cardiomyocytes [35,36,37,38,47,48,49,50,51].

We have previously used a multidisciplinary approach to understand the molecular mechanism for how the CaM E105A mutation, reported in a young individual who experienced an aborted cardiac arrest, leads to severe cardiac pathology [27,37]. More specifically, this de novo missense mutation c.A314>C in exon 5 of the *CALM1* gene was identified in a seemingly healthy 6-year-old boy, with no family history of cardiac disease, who exhibited profound QT prolongation with an increasing heart rate before the recurrence of polymorphic ventricular tachycardia at a pediatric intensive care unit [27]. In this CaM mutant, a glutamic acid (E) at 105 position within the third EF-hand motif of CaM, which crystallographic data suggests is directly involved in the Ca^2+^-binding of CaM C-domain, is substituted by an alanine (A) residue (Figure 5). We showed that expression of CaM^E105A^ mutant in zebrafish resulted in cardiac arrhythmia and increased heart rate [37]. In addition, while circular dichroism (CD) and thermal denaturation experiments suggested a deleterious effect of the E105A mutation on CaM stability, experiments on Ca^2+^-binding affinities of N- and C-lobes of the CaM^E105A^ mutant revealed a ~10-fold reduction of binding affinity of CaM^E105A^ C-lobe for Ca^2+^ compared to CaM^WT^ consistent with the molecular modeling structural predictions (Figure 5). Furthermore, co-immunoprecipitation experiments of native RyR2 from pig cardiac SR with CaM^WT^ and CaM^E105A^, suggested that the E105A mutation leads to a dramatic decrease in RyR2-CaM binding (over ~70%) at all Ca^2+^ concentrations. These findings were in accord with [^3^H]ryanodine binding assays, which revealed that inhibition of ryanodine binding to RyR2 by the CaM^E105A^ variant was almost abolished at all high Ca^2+^ concentrations, suggesting a significantly negative impact of CaM E105A mutation on CaM-RyR2 association [37]. Interestingly, recent ITC experiments led to the proposal that two regions of human RyR2 (3584-3602aa and 4255-4271aa) might contribute to a putative intra-subunit CaM-binding pocket indicating that some arrhythmogenic CaM mutations alter the interactions of CaM with this RyR2 CaM-binding pocket through differential molecular mechanisms [38].

The findings of this study strongly support our previously published work. Firstly, they concur with our findings regarding the deleterious effect of the E105A mutation on CaM’s association with RyR2. Secondly, and more importantly, our current data highlight the key role of two human RyR2 regions, 3584-3602aa, and 4255-4271aa, which potentially contribute to a putative intra-subunit RyR2 CaM-binding pocket that may be essential for successful CaM/RyR2 association and thus channel regulation (Figure 6).

Although the sequence that lies within the residues 3583-3603aa in RyR2 has been identified from multiple studies as the main and most well-established CaMBD region of RyR2 [16,40,41,42], up-to-date many other linear fragments of RyR2 have also been reported as potential CaM-binding sequences but their exact role is not yet clear. Another study that also used ITC to investigate the binding of CaM to three distinct mouse RyR2 regions (1941-1965aa, 3580-3606aa, and 4246-4276aa) showed that all three RyR2-regions studied were bound to CaM, but the 1941-1965aa region had much lower affinity for CaM compared to the other two RyR2 regions [54]. Interestingly, the contribution of the two aforementioned RyR2 regions to an intra-subunit RyR2 CaM-binding pocket is supported by another study that employed cryo-EM and FRET experiments to investigate the 3581-3612aa and 4261-4286aa regions of mouse RyR2 as putative CaMBDs [55]. These authors reported that upon RyR2 activation the structural domains bearing these two sequences move apart, indicating that conformational changes associated with channel activation occur in the vicinity of receptor-bound CaM, and these conformational changes affect CaM binding and vice versa, and thus the channel gating [55]. Furthermore, a recent crystallographic study revealed that the binding of CaM to RyR2 4246-4275aa region, and the binding of a fifth Ca^2+^ to CaM, are important for the physiological regulation of the RyR2 channel. It was demonstrated that the binding of the fifth Ca^2+^ to CaM results in a 2-fold increase in the binding affinity of the CaM-RyR2 complex, which might be critical for the stabilization of the CaM-RyR2 complex under physiological conditions [56].

The de novo missense CaM E105A mutation that was identified in exon 5 of the *CALM1* gene in a 6-year-old boy was associated with an LQTS phenotype [27]. LQTS is a life-threatening arrhythmia syndrome, in which repolarization of the heart after the heartbeat is affected, giving rise to an abnormally lengthy QT interval. This is primarily caused by dysfunction of the sarcolemmal voltage-gated Na^+^, Ca^2+^, and K^+^ channels that control the action potential [41]. In contrast to LQTS, CPVT is another arrhythmogenic disorder, which is characterized by adrenergic-induced bidirectional and polymorphic VT, which can lead to syncope or sudden cardiac death. CPVT has been associated with mutations in RyR2 and other accessory proteins, such as CSQ2 and CaM [6,28,41,57,58].

Interestingly, all our previous reports on the characterization of various arrhythmogenic CaM mutants support the notion that the clinical presentation of LQTS or CPVT associated with CaM mutations may involve a combination of both altered intrinsic Ca^2+^-binding to CaM, as well as a defective interaction of CaM with RyR2 that leads to defective regulation of this channel [35,36,37,38]. The physiological regulation of RyR2 Ca^2+^ release is tightly controlled, and a defective inhibition of the open probability of this channel might be a common factor of both LQTS and CPVT arrhythmias caused by CaM mutations, resulting in abnormal RyR2 Ca^2+^ release. However, in other instances, the defective RyR2 regulation by CaM might not be the primary cause of arrhythmogenesis, rather the significantly altered Ca^2+^-binding affinities of some arrhythmogenic CaM mutants could have a dominant impact due to the dysfunction of other ion channels within the cardiomyocytes. This is further supported by the identification of CaM mutations in patients with clinical presentations of mixed LQTS and CPVT phenotypes (e.g., the previously reported CaM D132E and Q136P mutations) [26], highlighting the fact that multiple factors can lead to defects in vital functions and interactions of this multifunctional Ca^2+^ sensor with RyR2 and the other ion channel complexes in the heart [36,37].

A step forward would be the resolution of the atomic level structure of the CaM^E105A^ mutant, as well as of other arrhythmogenic CaM mutants bound to the aforementioned RyR2 peptides that we suggest that contribute to a distinct mobile, intra-subunit calmodulin-binding domain. In conclusion, further investigation should help to delineate the complex molecular pathways whereby CaM mutations lead to life-threatening arrhythmogenic cardiac disease and to establish the mechanistic links between individual CaM variants and disease pathogenesis. This will help clinicians to select the optimal antiarrhythmic treatment in order to improve the survival of high-risk patients. In addition, systematic genetic screening for *CALM* variants should be essential for young individuals presented with arrhythmogenic cardiac disorders that will assist the development of future therapeutic strategies through precision medicine.

## 4. Materials and Methods

### 4.1. Plasmid Construction

The two CaM constructs (CaM^WT^ and CaM^E105A^) that were used in this study for recombinant protein production were the same as previously described [35,36,37]. Briefly, the human CaM^WT^ (GenBank^®^ accession number AAD45181.1) construct in pHSIE plasmid vector was subjected to oligonucleotide-mediated, site-directed mutagenesis (QuikChange II; Stratagene, La Jolla, CA, USA) to generate the CaM^E105A^ mutant in pHSIE vector [37].

### 4.2. Protein Expression and Purification

Recombinant CaM^WT^ and CaM^E105A^ proteins were expressed and purified as previously described [37,38]. For protein expression, *Escherichia coli* (BL21-CodonPlus(DE3); ThermoFisher) cells were transformed with the appropriate pHSIE-CaM plasmid and cultured at 37 °C until the A_600_ nm reached 0.6. Then protein expression was induced for 18 h at 16 °C with 0.1 mM IPTG (isopropyl β-d-thiogalactopyranoside), (Sigma-Aldrich, Burlington, MA, USA). The bacterial cell pellets were harvested by centrifugation at 6000× *g* for 15 min at 4 °C. Then, following lysis of the bacterial cells using lysozyme and 3 mild sonication cycles on ice, the recombinant CaM proteins were purified by one-step affinity chromatography purification, as previously described [37,38]. Finally, the eluted CaM proteins were dialyzed and concentrated using centrifugal concentrators (Sartorius, Gottingen, Germany; 3000 molecular weight cut-off), analyzed by SDS-PAGE and immunoblot analysis, as previously described [37,38] and the recombinant proteins were used for the ITC experiments. Peptides that were used in this study were synthesized by GenScript, Piscataway, NJ, USA.

### 4.3. Isothermal Titration Calorimetry

ITC studies were performed on a Nano ITC (TA Instruments, New Castle, DE, USA) microcalorimetry system at 25 °C, under both Ca^2+^-saturated and Ca^2+^-free conditions. Ca^2+^-saturated solutions were prepared in 100 mM KCl, 10 mM HEPES (pH 7.4), 10 mM CaCl_2_ buffer (holo-buffer), while Ca^2+^-free solutions were prepared in 100 mM KCl, 10 mM HEPES (pH 7.4), 10 mM EDTA buffer (apo-buffer). All sample solutions in the selected buffers were thoroughly degassed before performing ITC titrations at room temperature. For these binding assays, the calorimetric cell was filled with 18 μM of protein sample (CaM), while the syringe was loaded with a 200 μM peptide solution, corresponding to human RyR2 regions 3584-3602aa (peptide B) or 4255-4271aa (peptide F) in each case. The titration sequence included an initial 1 μL injection, followed by 15 identical 2.5 μL injections at 300s intervals, under a constant stirring speed of 350 rpm to ensure the rapid equilibration of the mixture. Separate blank experiments were performed to subtract the dilution heat of each component from the final binding isotherm. All ITC data were processed using the NanoAnalyze data analysis software (version 3.11.0—TA Instruments, New Castle, DE, USA) and plotted using the Microcal Origin 2015 software (v.9.2, January 2015) (OriginLab, Northampton, MA, USA). The stoichiometry of the interaction (moles of peptide bound per mol of protein) (N), the binding constant (K_b_ = 1/K_d_), and the molar binding enthalpy (Δ_r_H) of the reaction are obtained, along with their corresponding uncertainties, directly from the non-linear least squares fitting of a one set-of-sites model to the experimental data. The Gibbs free energy change upon binding (Δ_r_G) and the entropy change (Δ_r_S) accompanying the complex formation are calculated from the equations:Δ_r_G = RT ln K_b_ = Δ_r_H − T Δ_r_S 
where R is the gas constant, and T is absolute temperature. The uncertainties of Δ_r_G and Δ_r_S are estimated using error propagation calculations.

### 4.4. Molecular Modeling

Molecular operating environment software (MOE version 2022.02—Molecular Operating Environment (MOE), 2022.02 Chemical Computing Group ULC, Montreal, QC, Canada, 2023) was used to create the E105A residue mutation and the final energy minimized structures. After inserting the mutation using the Protein Build tool, the structure was checked for missing atoms, bonds, and contacts, and then a 3D protonation and energy minimization was performed using an AMBER10:EHT force field. The final energy-minimized structures were aligned and overlaid using the Overlap tool of the software with no further adjustments.

## Figures and Tables

**Figure 1 ijms-24-15630-f001:**
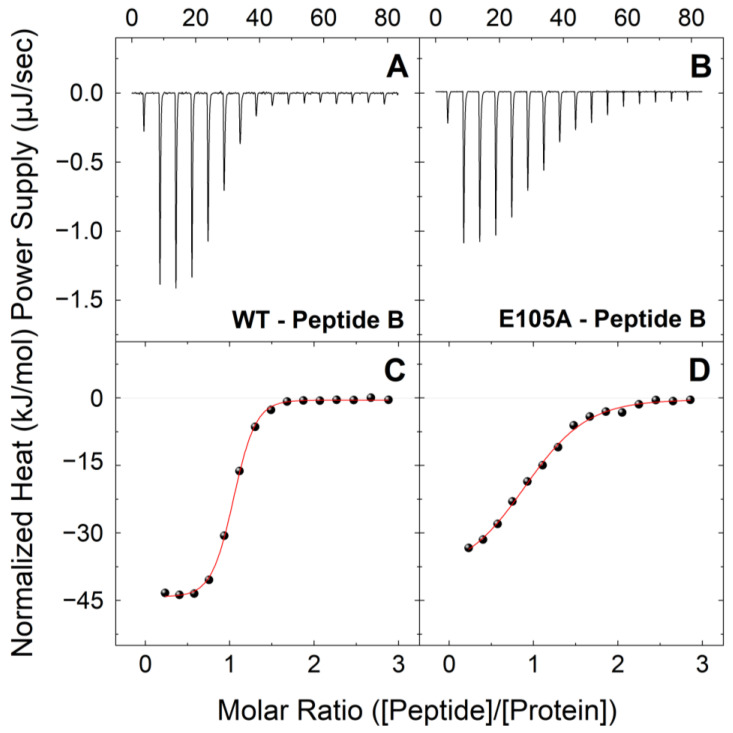
**Binding interactions of CaM^WT^ and CaM^E105A^ mutant with peptide B (RyR2 3584-3602aa) in holo-buffer.** Upper panels: Change in power supply to the calorimetric cell during the titration of 200 μM of a peptide B solution into 18 μM of CaM^WT^ (**A**) and CaM^E105A^ (**B**) at 25 °C in holo-buffer, after the subtraction of the appropriate reference experiments. Lower panels: Integration of the area under each injection, normalized per mol of injectant and plotted as a function of the (peptide)/(CaM) ratio at each point of the CaM^WT^ (**C**) and CaM^E105A^ (**D**) titrations. Solid red lines represent the non-linear least-square fit of the ITC data to a single-set-of-sites thermodynamic model.

**Figure 2 ijms-24-15630-f002:**
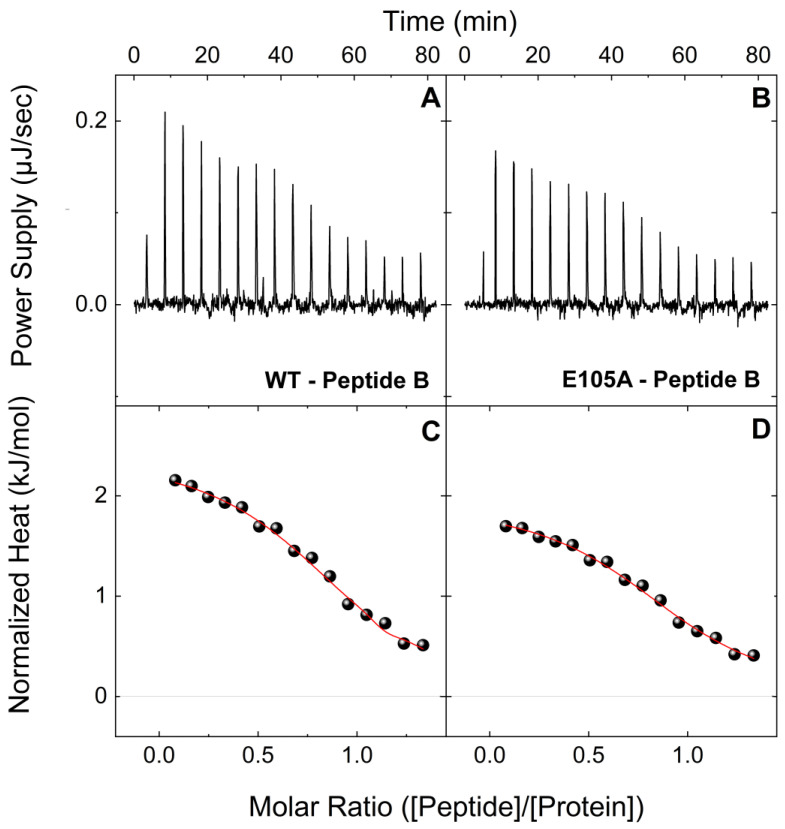
**Binding interactions of CaM^WT^ and CaM^E105A^ mutant with peptide B (RyR2 3584-3602aa) in apo-buffer.** Upper panels: Change in power supply to the calorimetric cell during the titration of 200 μM of a peptide B solution into 18 μM of CaM^WT^ (**A**) and CaM^E105A^ (**B**) at 25 °C in apo-buffer, after the subtraction of the appropriate reference experiments. Lower panels: Integration of the area under each injection, normalized per mol of injectant and plotted as a function of the (peptide)/(CaM) ratio at each point of the CaM^WT^ (**C**) and CaM^E105A^ (**D**) titrations. Solid red lines represent the non-linear least-square fit of the ITC data to a single-set-of-sites thermodynamic model.

**Figure 3 ijms-24-15630-f003:**
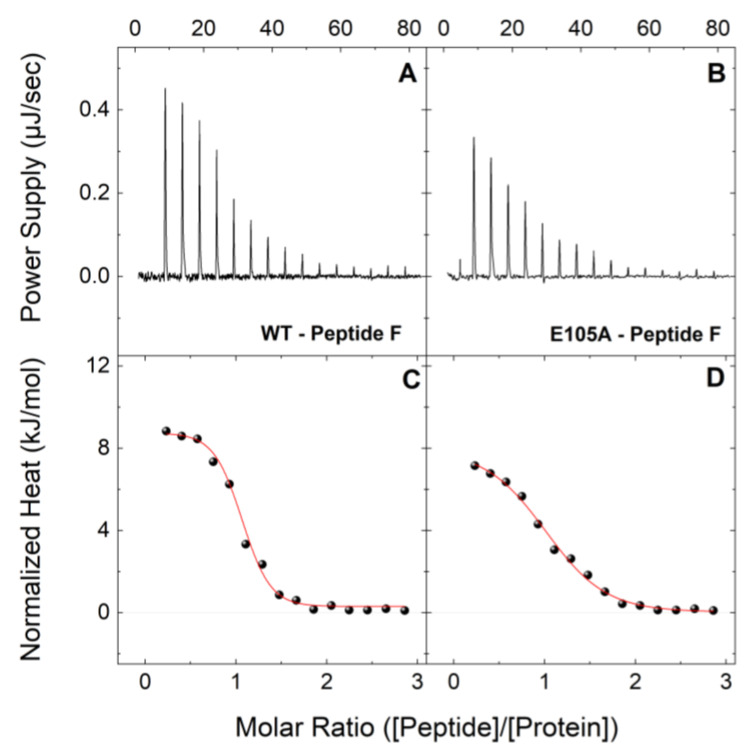
**Binding interactions of CaM^WT^ and CaM^E105A^ mutant with peptide F (RyR2 4255-4271aa) in holo-buffer.** Upper panels: Change in power supply to the calorimetric cell during the titration of 200 μM of a peptide B solution into 18 μM of CaM^WT^ (**A**) and CaM^E105A^ (**B**) at 25 °C in holo-buffer, after the subtraction of the appropriate reference experiments. Lower panels: Integration of the area under each injection, normalized per mol of injectant and plotted as a function of the (peptide)/(CaM) ratio at each point of the CaM^WT^ (**C**) and CaM^E105A^ (**D**) titrations. Solid red lines represent the non-linear least-square fit of the ITC data to a single-set-of-sites thermodynamic model.

**Figure 4 ijms-24-15630-f004:**
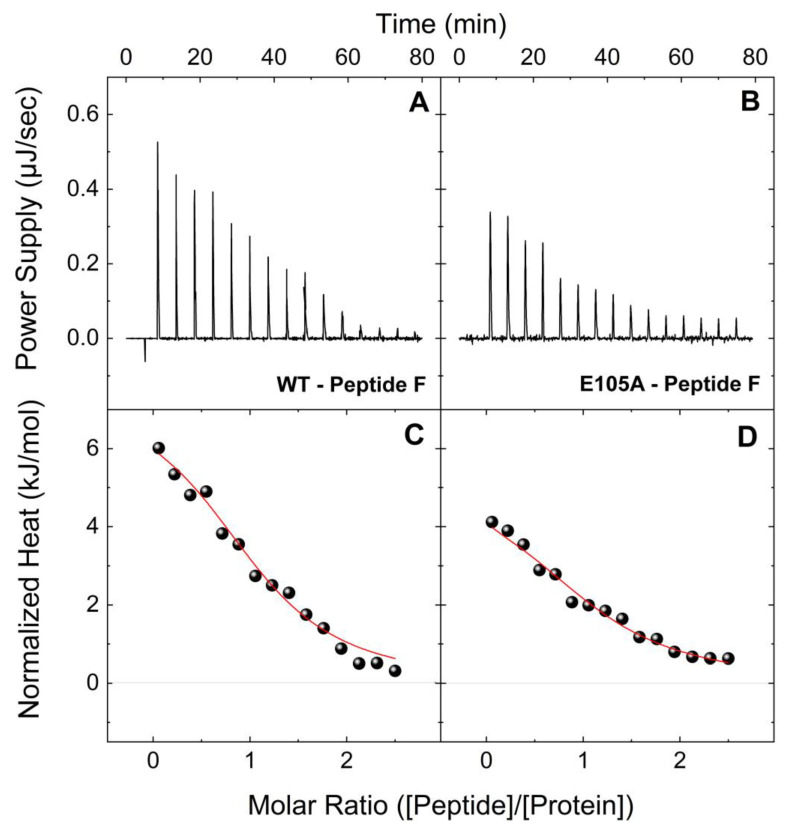
**Binding interactions of CaM^WT^ and CaM^E105A^ mutant with peptide F (RyR2 4255-4271aa) in apo-buffer.** Upper panels: Change in power supply to the calorimetric cell during the titration of 200 μM of peptide F solutions into 18 μM of CaM^WT^ (**A**) and CaM^E105A^ (**B**) at 25 °C in apo-buffer, after the subtraction of the appropriate reference experiments. Lower panels: Integration of the area under each injection, normalized per mol of injectant and plotted as a function of the (peptide)/(CaM) ratio at each point of the CaM^WT^ (**C**) and CaM^E105A^ (**D**) titrations. Solid red lines represent the non-linear least-square fit of the ITC data to a single-set-of-sites thermodynamic model.

**Figure 5 ijms-24-15630-f005:**
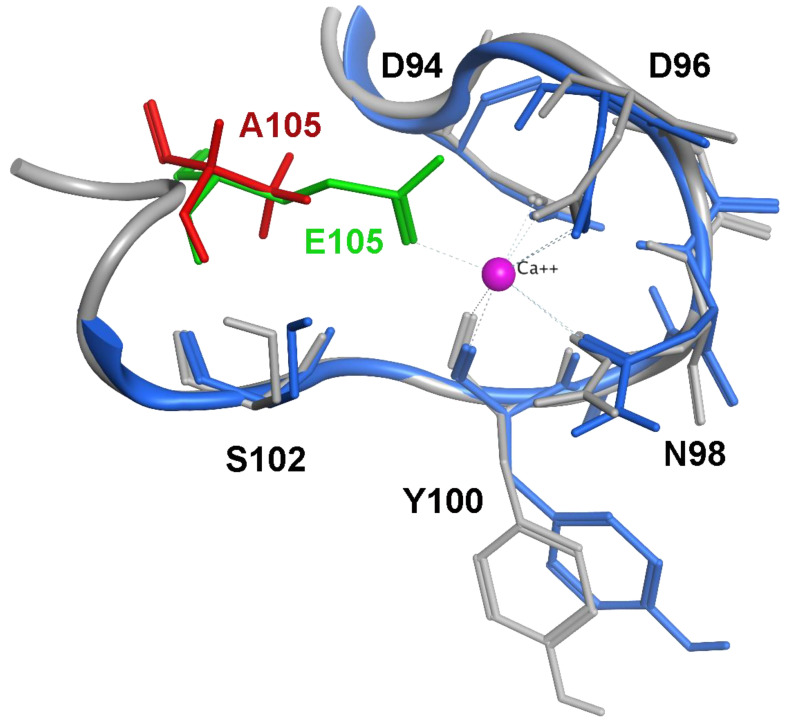
Overlap of the energy-minimized structures of CaM^WT^ (grey ribbon) and CaM^E105A^ (blue ribbon) at the EF-hand III binding site. Important residues are shown as stick models, with E105 and A105 colored green and red, respectively, for reasons of emphasis. The model is based on the PDB entry with code 1CLL [52].

**Figure 6 ijms-24-15630-f006:**
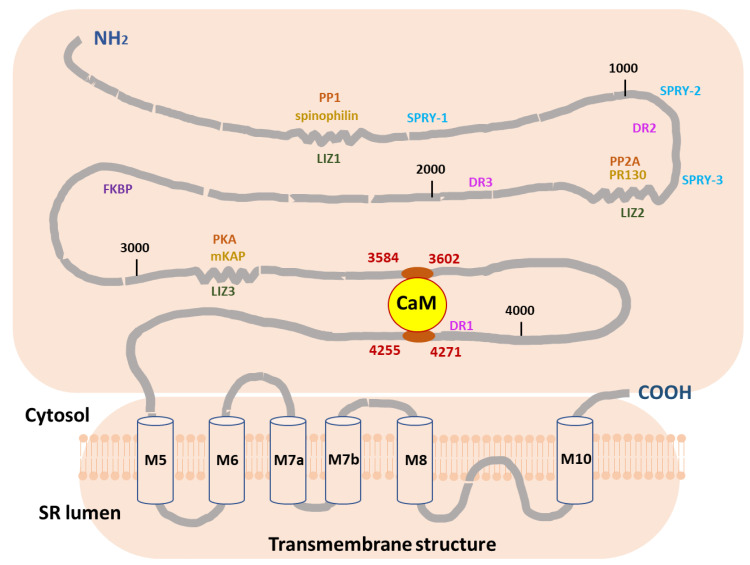
**Proposed mechanism of RyR2 regulation by CaM.** The figure demonstrates the two human RyR2 CaM-binding regions (3584-3602aa and 4255-4271aa), which we propose contribute to a potential intra-subunit RyR2 CaM-binding pocket, essential for CaM/RyR regulation. The distribution of other satellite proteins across structural domains of RyR2 is also presented. CaM, Calmodulin; DR, divergent region; FKBP, calstabin 2; LIZ, leucine–isoleucine zipper; PKA, protein kinase A; PP, protein phosphatase; SPRY; mKAP, PR130; PP1, phosphatase 1; PP2A, phosphatase 2A; RyR2, ryanodine receptor 2 (modified from [53]).

**Table 1 ijms-24-15630-t001:** Thermodynamic parameters for the binding of peptide B (RyR2 3584-3602aa) with CaM^WT^ and CaM^E105A^ mutant in holo- and apo-buffer. Dissociation constant (K_d_), binding enthalpy change (Δ_r_H), entropic term change (−T∙Δ_r_S), and free energy change (Δ_r_G) for the interaction of peptide B with CaM^WT^ and CaM^E105A^ mutant samples at T = 25 °C in 100 mM KCl, 10 mM HEPES (pH 7.4), 10 mM CaCl_2_ buffer (holo-buffer) and 100 mM KCl, 10 mM HEPES (pH 7.4), 10 mM EDTA (apo-buffer). Values and corresponding errors were derived from non-linear least square fit of the ITC data to a one-set-of-sites thermodynamic model.

Titration		Dissociation Constant (K_d_) (μM)	Stoichiometry [N]	Binding Enthalpy (Δ_r_H) (kJ/mol)	Entropic Term (−T∙Δ_r_S) (kJ/mol)	Gibbs Free Energy Change (Δ_r_G) (kJ/mol)
**Peptide B**	**Holo-CaM^WT^**	0.41 ± 0.05	0.95 ± 0.01	−44.5 ± 1.8	8.1 ± 1.8	−36.5 ± 0.3
	**Holo-CaM^E105A^**	2.29 ± 0.24	0.93 ± 0.01	−34.2 ± 1.5	2.0 ± 1.5	−32.2 ± 0.3
**Peptide B**	**Apo-CaM^WT^**	4.54 ± 0.51	0.91 ± 0.01	2.3 ± 0.1	−32.9 ± 0.3	−30.5 ± 0.3
	**Apo-CaM^E105A^**	7.04 ± 1.02	0.87 ± 0.01	1.9 ± 0.6	−31.3 ± 0.4	−29.4 ± 0.4

**Table 2 ijms-24-15630-t002:** Thermodynamic parameters for the binding of peptide F (RyR2 4255-4271aa) with CaM^WT^ and CaM^E105A^ mutant in holo- and apo-buffer. Dissociation constant (K_d_), binding enthalpy change (Δ_r_H), entropic term change (−T∙Δ_r_S), and free energy change (Δ_r_G) for the interaction of peptide F with CaM^WT^ and CaM^E105A^ mutant samples at T = 25 °C in 100 mM KCl, 10 mM HEPES (pH 7.4), 10 mM CaCl_2_ buffer (holo-buffer) and 100 mM KCl, 10 mM HEPES (pH 7.4), 10 mM EDTA (Apo Buffer). Values and corresponding errors were derived from non-linear least square fit of the ITC data to a one-set-of-sites thermodynamic model.

Titration		Dissociation Constant (K_d_) (μM)	Stoichiometry (N)	Binding Enthalpy (Δ_r_H) (kJ/mol)	Entropic Term (−T∙Δ_r_S) (kJ/mol)	Gibbs Free Energy Change (Δ_r_G) (kJ/mol)
**Peptide F**	**Holo-CaM^WT^**	0.63 ± 0.07	0.97 ± 0.01	9.1 ± 0.5	−44.5 ± 0.7	−35.4 ± 0.3
	**Holo-CaM^E105A^**	2.58 ± 0.28	0.94 ± 0.01	7.8 ± 0.4	−39.7 ± 0.5	−31.9 ± 0.3
**Peptide F**	**Apo-CaM^WT^**	11.05 ± 1.36	0.93 ± 0.01	6.6 ± 0.5	−34.9 ± 0.6	−28.3 ± 0.3
	**Apo-CaM^E105A^**	24.48 ± 3.29	0.89 ± 0.01	5.2 ± 0.5	−31.6 ± 0.6	−26.3 ± 0.3

## Data Availability

Data is contained within the article.

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
