# Peer review of "Arrhythmia-Associated Calmodulin E105A Mutation Alters the Binding Affinity of CaM to a Ryanodine Receptor 2 CaM-Binding Pocket"

_ijms, 2023, doi:10.3390/ijms242115630_

Round 1

Reviewer 1 Report

Comments and Suggestions for Authors

In this paper the author's apply isothermal titration calorimetry (ITC) to show that two regions in RyR2 binding with CaM strongly.  They show further thatthis binding affinity is reduced in the presence of the E105A mutation.

Overall, this is a solid study.   I feel that the author's can clarify a few points bellow to make the paper stronger:

1.  The results and discussion heading is out of place.

2.  line 129.   Increased heart rate is generally not related to tachycardia.  During tachycardia there is a periodic beating of the heart due to a reentrant electrical excitation.  However, this only lasts for a few periods.   This is not a fast heart rate, which is determined by the SA node.  Please review the basic literature on VT and heart rate abnormalities.

3.  Inspection of Table I and II shows that the binding enthalpy of Peptide B with holo-CaM wt is substantially different than all the other cases.   This implies that this binding process is fundamentally different.  For examply it is clear that this reaction is driven primarily by energetic processes rather than entropic.  This is not the case for the 3 other reaction. 

I think the large discrepancy between this case and all the rest deserves further discussion.

4.  I believe this paper should have a more extensive discussion about the author's previous work.  For example, the proposed mechanism shown in Figure 6 assumes that the the binding pocket is formed by the two RyR2 regions in question.   However, is this consistent with the latest Cryo-EM structures of RyR2?   For example, it would be very informative to see the location of these two regions in the full crystal structure.  This structure is now well known.   

Author Response

1. The results and discussion heading is out of place.

We would like to thank the reviewer for his very positive comments, his time and suggestions to further improve our manuscript. We have now separated the Results and Discussion sections as per the reviewer's suggestions and we have added sub-headings wherever it was necessary.

2. line 129.   Increased heart rate is generally not related to tachycardia.  During tachycardia there is a periodic beating of the heart due to a reentrant electrical excitation.  However, this only lasts for a few periods.   This is not a fast heart rate, which is determined by the SA node.  Please review the basic literature on VT and heart rate abnormalities.

We thank the reviewer for this note. We have now removed the phrase “suggestive to ventricular tachycardia”.

3. Inspection of Table I and II shows that the binding enthalpy of Peptide B with holo-CaM wt is substantially different than all the other cases.   This implies that this binding process is fundamentally different.  For examply it is clear that this reaction is driven primarily by energetic processes rather than entropic.  This is not the case for the 3 other reaction. 

I think the large discrepancy between this case and all the rest deserves further discussion.

We thank the reviewer for this note. We would like to clarify that generally negative enthalpies indicate an extensive network of interactions between peptides and proteins, while endothermic events mainly describe binding processes that dominated by hydrophobic interactions. These differences in enthalpy not necessarily suggest a different binding mechanism. Similar effect has also been previously reported with other CaM binding peptides [1, 2]. We have also added the below to our results section:

"The endothermic binding signature of peptide F-CaM complex indicates a binding event, which is stabilized mainly by electrostatic interactions and hydrophobic effects. In contrst, the binding of CaM to peptide B in the presence of Ca2+ shows an exothermic ITC thermogram indicating favorable interactions of peptide-protein polar groups. In the absence of Ca2+ the total charge of the protein is altered giving a binding curve that is governed mainly by electrostatic interactions."

  1. Chang, B. J., A. B. Samal, J. Vlach, T. F. Fernandez, D. Brooke, P. E. Prevelige, Jr., and J. S. Saad. "Identification of the Calmodulin-Binding Domains of Fas Death Receptor." PLoS One 11, no. 1 (2016): e0146493.
  2. Chavan, T. S., S. Abraham, and V. Gaponenko. "Application of Reductive (1)(3)C-Methylation of Lysines to Enhance the Sensitivity of Conventional Nmr Methods." Molecules 18, no. 6 (2013): 7103-19.

4. I believe this paper should have a more extensive discussion about the author's previous work.  For example, the proposed mechanism shown in Figure 6 assumes that the the binding pocket is formed by the two RyR2 regions in question.   However, is this consistent with the latest Cryo-EM structures of RyR2?   For example, it would be very informative to see the location of these two regions in the full crystal structure.  This structure is now well known.   

We do appreciate reviewer’s suggestion; however, the Cryo-EM structure of RyR2 is missing large segments of the protein including the two RyR2 regions (3584-3602aa and 4255-4271aa) that we investigated in the current study. This is not surprising as these chains are highly flexible to regulate calcium influx and cryo-M is still a low-resolution technique. Hopefully, higher resolution structures in the future may allow a more detailed analysis to combine and confirm our experimental results with structural data. In our discussion though, we have included a previous study that used cryo-EM and FRET experiments to investigate the equivalent RyR2 regions  of mouse RyR2 (3581-3612aa and 4261-4286aa) and it is supportive to our hypothesis.

“…Interestingly, the contribution of the two aforementioned RyR2 regions to an intra-subunit RyR2 CaM-binding pocket is supported by another study that employed cryo-EM and FRET experiments to investigate the 3581-3612aa and 4261-4286aa regions of mouse RyR2 as putative CaMBDs. These authors reported that upon RyR2 activation the structural domains bearing these two sequences move apart, indicating that conformational changes associated with channel activation occur in the vicinity of receptor bound CaM, and these conformational changes affect CaM binding and vice versa, and thus the channel gating...”

Huang, X., Y. Liu, R. Wang, X. Zhong, Y. Liu, A. Koop, S. R. Chen, T. Wagenknecht, and Z. Liu. "Two Potential Calmodulin-Binding Sequences in the Ryanodine Receptor Contribute to a Mobile, Intra-Subunit Calmodulin-Binding Domain." J Cell Sci 126, no. Pt 19 (2013): 4527-35.

Reviewer 2 Report

Comments and Suggestions for Authors

Author Response

1. The Results and Discussion section should be divided into two separate sections. The Results section should present the experimental data and should not contain any interpretation or speculative information. The separate Discussion section can then present the interpretation of the data and explain what the data means. For example, the presentation and description of the ITC experiments and data should appear in the Results. The speculation about connecting the two peptide fragments into a single site (Fig. 6) should be presented in the Discussion. That way it will be easier to distinguish experimental facts from interpretation and not be confused by what is known and what is speculation.

We would like to thank the reviewer for his positive comments and his time to review and improve our manuscript. We have now separated the Results and Discussion sections, as per the reviewer's suggestions and we have added sub-headings wherever it was necessary. As per reviewer’s suggestion, Figure 6 is now presented on the Discussion section.

2. The E105A mutation has a much larger effect on the Ca2+-bound CaM binding to each peptide compared to the much smaller effect on apoCaM binding. Also, the apoCaM binding (even for wild type) is very low affinity (10 micromolar) and is outside the physiological concentration range of apoCaM in cardiac cells. Therefore, the apoCaM binding to RyR2 is probably not physiologically relevant and only the Ca2+-bound CaM is capable of binding to the peptides under physiological conditions. The simplest interpretation of the ITC data is that the E105A mutation is probably disabling Ca2+ binding to the 3rd EF-hand, which causes this EF-hand to adopt a Ca2+-free closed conformation (even in the presence of Ca2+) that has lower binding affinity for the peptide, akin to that of apoCaM. To test this idea, it would be interesting to solve the atomic-level structure of E105A bound to the peptide. Also, this idea should be added to the Discussion to discuss the role of Ca2+ binding and the effect of the E105A mutation on Ca2+ binding etc.

We thank the reviewer for his suggestion. We have mentioned in our discussion that experiments on Ca2+-binding affinities of N- and C-lobes of the CaME105A mutant revealed a ~10-fold reduction of binding affinity of CaME105A C-lobe for Ca2+ compared to CaMWT, which is also consistent with the molecular modelling structural predictions that we show in Figure 5 (highlighted text). However, we would like to clarify that our ITC experiments have been performed at saturated Ca2+ conditions, where all four site of CaM are occupied. Thus, even if there is a lower affinity for Ca2+ in the 3rd EF hand domain, it is not Ca2+ free and the change in the binding affinity with the peptide is attributed to the differences in electrostatic interactions due to the substitution of the charged amino acid E (glutamic acid) to the neutral A (alanine) and the changes in the Ca2+ coordination that this might cause. We do agree with the reviewer that a step-forward is to solve the atomic-level structure of E105A bound to the peptide, which we have added on the discussion.

“…experiments on Ca2+-binding affinities of N- and C-lobes of the CaME105A mutant revealed a ~10-fold reduction of binding affinity of CaME105A C-lobe for Ca2+ compared to CaMWT consistent with the molecular modelling structural predictions.”

“A step forward would be the resolution of the atomic level structure of the CaME105A mutant, as well as of other arrhythmogenic CaM mutants bound to the aforementioned RyR2 peptides that we suggest that contribute to a distinct mobile, intra-subunit calmodulin-binding domain.”

3. Does CaM bind to full-length RyR2 with the same affinity and stoichiometry as its binding to the peptides? The CaM binding to the simple peptides seems artificial and may not mimic CaM binding to the full-length RyR2. This paper needs additional experiments or data to demonstrate that CaM binding to the peptides is physiologically relevant and mimics CaM binding to the full-length RyR2. For example, do the peptides serve as a potent competitive inhibitor of RyR2 channel function? Does CaM bind to full-length RyR2 with the same affinity as the peptide? If you delete the two peptide regions in RyR2, does the deletion abolish Ca2+-dependent channel inactivation or channel function?

We would like to thank the reviewer and to point out that these peptides correspond to well established RyR2 calmodulin binding regions. Several previous studies have provided significant physiological evidence regarding the importance of the RyR2 3584-3602aa region as the main CaM binding RyR2 region. For example, deletion of the sequence 3583-3603aa in RyR2 resulted in disruption of CaM binding to RyR2 and reduced efficacy of CaM inhibition on RyR2 activity in single channel measurements [1]. In addition, the functional effect of CaM has been analyzed on a series of chimeras, deletion and point mutations within this region. Introduction of mutations into this RyR2 region has been shown to cause early cardiac hypertrophy in mice and in neonatal cardiomyocytes, as well as an impaired inhibition by CaM [2]. Such mutation have not yet been performed for the 4255-4271aa region but recent crystallographic study revealed the binding of CaM to RYR2 4246-4275aa region as well as the binding of a fifth Ca2+ to CaM, which may contribute to the physiological regulation of RyR2 channel. It was demonstrated that the binding of the fifth Ca2+ to CaM results in a 2-fold increase in the binding affinity of CaM-CaMBD3 complex, which might be critical for the stabilization of the CaM-RyR2 complex under physiological conditions [3].

  1. Yamaguchi, N., L. Xu, D. A. Pasek, K. E. Evans, and G. Meissner. "Molecular Basis of Calmodulin Binding to Cardiac Muscle Ca(2+) Release Channel (Ryanodine Receptor)." J Biol Chem 278, no. 26 (2003): 23480-6.
  2. Yamaguchi, N., N. Takahashi, L. Xu, O. Smithies, and G. Meissner. "Early Cardiac Hypertrophy in Mice with Impaired Calmodulin Regulation of Cardiac Muscle Ca Release Channel." J Clin Invest 117, no. 5 (2007): 1344-53.
  3. Yu, Q., D. E. Anderson, R. Kaur, A. J. Fisher, and J. B. Ames. "The Crystal Structure of Calmodulin Bound to the Cardiac Ryanodine Receptor (Ryr2) at Residues Phe4246-Val4271 Reveals a Fifth Calcium Binding Site." Biochemistry 60, no. 14 (2021): 1088-96.

Round 2

Reviewer 2 Report

Comments and Suggestions for Authors

The revised manuscript addresses my previous concerns.